# Coupling Coordination Degree between the Socioeconomic and Eco-Environmental Benefits of Koktokay Global Geopark in China

**DOI:** 10.3390/ijerph19148498

**Published:** 2022-07-12

**Authors:** Yiting Zhu, Xueru Pang, Chunshan Zhou, Xiong He

**Affiliations:** 1Key Laboratory of Sustainable Development of Xinjiang’s Historical and Cultural Tourism, College of Tourism, Xinjiang University, Urumqi 830049, China; yiting@xju.edu.cn (Y.Z.); pxr@stu.xju.edu.cn (X.P.); 2School of Geography and Planning, Sun Yat-sen University, Guangzhou 510275, China; hexiong6@mail2.sysu.edu.cn

**Keywords:** socioeconomic benefits, eco-environmental benefits, coupling coordination degree, Koktokay Global Geopark

## Abstract

The rapid economic growth of geoparks has put pressure on their ecological environments. Therefore, to ensure the sustainable development of geoparks, we must explore the coupling relationship between their socioeconomic benefits (SEBs) and eco-environmental benefits (EEBs). Based on coupling coordination theory and using statistical data from 2005 to 2018, in this study, we aimed to establish an indicator system for evaluating the coupling coordination degree (CCD) between the SEBs and EEBs of the Koktokay Global Geopark in China, which is both theoretically and practically relevant for research on the sustainable development of geoparks. As a result, we found the following: First, the comprehensive development level of the SEBs of the Koktokay Global Geopark showed a fluctuating upward trend during the study period. Second, the comprehensive development level of the EEBs of the geopark remained stable but fluctuated slightly: it declined from 2009 to 2012, affected by the deterioration of the eco-environment, and fell to its lowest point in 2012. By strengthening the protection of the eco-environment of geoparks, the EEBs gradually improved and became stable. Finally, we found that the CCD between the SEBs and EEBs of the Koktokay Global Geopark improved from mildly disordered to basically coordinated, indicating that the CCD is developing toward an increasingly higher level. The purpose of this study was to promote the reasonable development of geotourism while focusing on a sound eco-environment and to provide recommendations for the sustainable development of the Koktokay Global Geopark and a reference for the development of other similar geoparks.

## 1. Introduction

Geoparks were first proposed by UNESCO in the UNESCO Geological Parks Program, which mainly focuses on geological heritage. A park is a place with economic attributes and provides public service and recreation functions. Geoparks are territories with unique geological heritage and sustainable territorial development [1]. They contribute to the growth of local economies through sustainable tourism, and provide education and protection services [2]. For example, China’s Dunhuang UNESCO Global Geopark, located in the Belt and Road area, is rich in geological heritage. Because the local administration and residents have taken effective protection actions and widely conveyed geoscience knowledge and regional communities have actively participated in geotourism activities, the awareness of residents and tourists in protecting geological heritage and the eco-environment has been improved, thereby promoting the sustainable development of tourism [3]. Geoparks can be divided into National and Global Geoparks. UNESCO officially approved the International Geoscience and Geoparks Programme (IGGP) in 2015, which integrated all existing Global Geoparks into the UNESCO Global Geoparks plan, and advocated for the protection of geological heritage and the sustainable development of geoparks. UNESCO Global Geoparks are not only single and unified areas with geological heritage of international importance but also living territories where stakeholders work together to construct a sustainable future [4].

Geoparks are different from regular parks, having their own unique characteristics. Geoparks contain precious geological heritage resources that have been formed and preserved by the evolution of the Earth over hundreds of millions of years, and have scientific, aesthetic, and educational value. The primary aim of geoparks is to protect their nonrenewable geological heritage resources, promote their geological heritage to the public, and generate new job opportunities for the local community by conducting geotourism related to their geological heritage, which provides real economic benefits to the local community and supports the sustainable development of geoparks [5,6,7]. In addition, one of the tasks of geoparks is to popularize geoscience knowledge and provide education and meet science aims by displaying unique geological heritage landscapes related to nature and culture [8], which is also the difference between geoparks and other regular parks or scenic spots. Therefore, tourists who attend geoparks not only want to enjoy the aesthetics of the landscapes but also want to obtain geoscience knowledge to enhance their understanding of the natural evolution process. Some tourists want to increase their knowledge and cultural experiences, and other tourists consider the acquisition of geoscience knowledge as part of a wider tourism experience [9,10]. Moreover, geoproducts are popular with tourists, which are produced by local communities. Many geoparks have developed their own geoproducts [11,12]. Geoproducts are commercial services or manufactured goods inspired by geodiversity that are innovative, new, or reinvented traditional products, including handicrafts, food, tourism facilities, etc. [13]. For example, the Hong Kong UNESCO Global Geopark [14], which provides tourists with geopark-themed dishes in restaurants, offers a geolicious menu that provides descriptions of the dish names, special plates, and their connections with the geopark’s features.

The most effective method to achieve geoconservation is to promote and explain it through geotourism [15]. Tourism has various socioeconomic benefits; geotourism, as a subcategory of tourism, can not only drive local economic benefits [16] but also improve the local community understanding of the value of geological heritage and strengthen their awareness of the importance of the protection of this heritage. Hence, a sustainable lifestyle can be stimulated to ensure areas become sustainable and resilient. The establishment of UNESCO Global Geoparks in 2015 has played an important role in achieving the UN Sustainable Development Goals. Through this program, geobrands have been created, the visibility of regions has increased, more tourists have been attracted, the pride of local communities has increased, and more relevant stakeholders have been encouraged to become involved in geopark activities [17]. Geoparks have been established all over the world to benefit local residents and tourists through geotourism. Currently, the number of UNESCO Global Geoparks totals 161 in 44 countries and regions around the world. China proposed building geoparks in 1985, with the original intention of protecting geological heritage. China is now the country with the largest number of Global Geoparks in the world, and their number is rapidly increasing. To date, China hosts 41 Global Geoparks, accounting for 25.5% of the global total and ranking first in the world. Since the proposal of the Belt and Road Initiative in 2013, countries along the Belt and Road have become the fastest-growing segment in terms of tourism revenue [18]; however, the environmental problems in the regions along the Belt and Road have attracted widespread attention. Most countries and regions along the Belt and Road are located in arid, semiarid, or semihumid areas with complex and fragile natural eco-environments that are fragile and sensitive and have weak self-recovery abilities [19]. Xinjiang is located in the Belt and Road region and has a diverse and vulnerable natural environment. Like most regions along the Belt and Road, Xinjiang is experiencing conflict between economic development and eco-environment protection. Therefore, Xinjiang is a region typical of the Belt and Road region that can be used for studying the ecosystem health and environmental geography. Additionally, the strata in Xinjiang, China are fully developed, preserving its evolutionary history from about 67 million to 4.6 billion years ago. The geological structure in Xinjiang is complex and diverse, the strata are fully exposed, and tectonic movement is frequent, which has created the characteristic pattern of topography and landforms in Xinjiang. Therefore, Xinjiang is an ideal area for the study of geoparks.

Geoparks are not only places for sightseeing, culture, and entertainment for the public but also key areas protecting geological heritage and the eco-environment. They are bases of scientific research and used for the popularization of geoscience knowledge, which play important roles in protecting geological heritage, conveying geoscience knowledge, and driving the development of tourism to promote the local economy. However, the rapid development of geoparks has not only promoted the development of the local economy but has also placed increased pressure on the eco-environment, resulting in downward trends in air and water quality, forest coverage, and the biodiversity index of some geoparks. The deterioration of the eco-environment has inhibited the development of tourism and reduced socioeconomic benefits, and is thus threatening the healthy and sustainable development of geoparks. Socioeconomic and eco-environmental systems are open systems that are coupled because of their rich values and complex structure. Coupling originates from physics; refers to the phenomenon in which two or more systems influence each other through various interactions; and is widely used in ecology, agriculture, economics, and other fields [20]. Coordination is the benevolent interrelationship among systems or system elements. As such, the coupling coordination degree (CCD) measures the degree of relationship and coordination of a system or system elements [21]. In this study, we considered the coupling and coordinated development of socioeconomic benefits (SEBs) and eco-environmental benefits (EEBs) in the pursuit of overall optimization and common development, and in synchronizing the development speed and direction of the two systems. Therefore, in the protection and development of geoparks, coordinating the relationship between their socioeconomic development and eco-environment is crucial for the sustainable development of geoparks.

The relationship between socioeconomic development and the eco-environment has not only attracted the attention of many countries [22,23] but its related research is also an increasingly popular and vital topic in sustainable development. In these studies, scholars have used a number of research methods, including macro-qualitative descriptions and quantitative methods, and have designed many models for evaluating the relationship between socioeconomic development and the eco-environment, of which the most representative is the environment Kuznets curve (EKC), which was proposed by Krueger [24]. The EKC reveals the U-shaped curve relationship between economic development and the eco-environment. For example, the EKC hypothesis was confirmed by studying the impact of agro-economic factors on greenhouse gas emissions in developing European economies in comparison with advanced European economies [25]. Researchers have analyzed the relationship between socioeconomic development and the eco-environment based on various models, such as the pressure–state–response (PSR) model [26,27], grey correlation degree model [28,29], system dynamics model [30], and so on. In addition, the methods often used to study the relationship between socioeconomic development and the eco-environment include CCD [31], geographical spatial analysis [32], regression analysis [33,34], weighted TOPSIS [35], etc. In terms of the research scale and area, the former has gradually changed from the macro to the micro scale [36], and the latter covers urban agglomeration [37], the provincial level [38], municipal level [39], county region [40], river basin level [41], etc. Many studies have been conducted on cities or urban agglomeration, mostly exploring the relationship between socioeconomic development and the eco-environment in the process of urbanization [42], or the relationship between urbanization and the eco-environment [21].

Currently, the bulk of research on the coupling coordination relationship between socioeconomic development and the eco-environment has mainly revolved around three aspects. The first point concerns the evaluation or analysis of their coupling coordination relationship. For example, some scholars have analyzed the coupling coordination development of economic development and the environment in China [43,44], whereas others have evaluated or analyzed the coupling coordination relationship of socioeconomic development and the eco-environment of a specific urban agglomeration [45], province [46], city [47], or basin [48]. The second aspect concerns the spatio-temporal characteristics of the coupling between socioeconomic development and the eco-environment. As shown in the literature, some researchers explored the spatio-temporal characteristics of the coupling coordination development of economic growth and environment in a region [49], whereas others focused on tourism, further exploring the spatio-temporal characteristics of the coupling coordination development of the tourism economy and eco-environment in a region [50]. The third aspect is the factors driving the coordinated development of socioeconomic development and the eco-environment. By analyzing the coupling coordination situation of socioeconomic development and the eco-environment in different research areas, scholars have summarized the factors driving their coupling coordination development. Landscape, terrain, traffic, and climate factors have important impacts on the coupling coordination relationship between the eco-environment and economy of counties in northern China [51]. Some scholars found that the annual average population and industrial wastewater discharge are the main factors contributing to the coupling coordination relationship between the economy and environment in resource-based cities. Resource-based cities usually rely on their urban resource endowments in their early economic development [52]. In addition, with the deepening and expansion of research, coupling analyses based on two, three [53], and four systems [54] have been conducted. Researchers have added logistics, energy, tourism, and other socioeconomic factors and eco-environment factors to their models. For example, an index system of economic development, logistics development, and the eco-environment was constructed by adding logistics factors, and the coupling degree model was used to calculate the CCD of 30 provinces and cities in China from 2008 to 2017 [55]. Energy factors were added to quantitatively study the coupling coordination relationship between energy, economy, and the eco-environment in Australia from 2007 to 2016 [56]. Some researchers focused on tourism, and explored the coupling coordination relationship between tourism and several other factors [57].

The aim of the establishment of geoparks is to protect geological heritage resources and promote the sustainable development of SEBs. The principle of “developing while protecting, protecting while developing” should be followed [58]. However, due to the fierce development that can characterize economic modernization, some geoparks have ignored the protection of geological heritage in pursuit of higher SEBs, resulting in a series of problems such as soil erosion, resource depletion, and environmental pollution, which have restricted the sustainable development of geoparks. Therefore, on the premise of protecting geological heritage, geoparks should fully take advantage of the value of geological heritage resources, reasonably develop geotourism, and work toward sustainable development, in which socioeconomic development and the eco-environment are coordinated. However, current studies on geoparks have mostly focused on geological heritage evaluations [59,60], geotourism [61,62], geological resource protection [63], geodiversity and biodiversity [64,65], the sustainable development of geoparks [66,67], etc. In particular, geological heritage evaluation and geotourism, as important aspects of achieving the sustainable development of geoparks, have attracted considerable attention from scholars in recent years. For example, the concepts of heritage and geodiversity were explained, and an inventory and numerical assessment method for geological heritage and geodiversity sites was proposed [68]. Additionally, the nature and characteristics of geotourism were defined, and the development of geotourism was discussed by referring to various cases [69]. The relationship between socioeconomic development and the eco-environment has not received enough attention in the studies of geoparks.

According to the above literature, the coupling coordination relationship between socioeconomic development and the eco-environment is related to sustainable development, with related research results becoming increasingly rich, and the protection and development of geoparks have attracted much attention. Regarding studies on the relationship between socioeconomic development and the eco-environment of geoparks, Yi et al. established an evaluation index system and evaluation model of the CCD of geoparks, and used Songshan Global Geopark as an example to test the rationality of the evaluation index system and model [70]. Few studies have focused on the coupling coordination relationship and evaluation of the socioeconomic development and eco-environment of geoparks. Coordinating the relationship between socioeconomic development and the eco-environment of geoparks has theoretical and practical value for the protection and sustainable development of geoparks. According to Yi et al. [70,71], constructing a more comprehensive and effective indicator system of the socioeconomic development and eco-environment of geoparks will meet not only the theoretical needs of sustainable geopark development research but also the practical needs of geopark managers and development. Therefore, using statistical data from 2005 to 2018, in this study, we adopted the entropy method and coupling coordination model to calculate the comprehensive development level of the SEBs and EEBs system of the Koktokay Global Geopark, discussed their variation characteristics, and identified the type of coupling coordination in this case. We developed an evaluation index system to assess the coupling coordination relationship between the SEBs and EEBs, which provides a theoretical reference for the development of the Koktokay Global Geopark. This study is valuable for the ecosystem health and sustainable development of environmental geography in the Belt and Road regions. Our aim was to find a more effective method for geoparks, especially geoparks in vulnerable environments, to balance eco-environmental protection and geotourism development, and to provide useful information for the sustainable development of geoparks.

## 2. Research Methods and Data Sources

### 2.1. Study Area

The study area was the Koktokay Global Geopark, located in Fuyun County and Qinghe County, Altay Region, Xinjiang Uygur Autonomous Region, China, in the inland area of central Asia. The Koktokay Global Geopark has a total area of 185,000 km^2^; its detailed location is shown in Figure 1. The Koktokay Global Geopark is recognized as the “Natural Geological Museum” by the global geological community and has beautiful natural and geological characteristics due to its unique diversity of natural minerals and strange rocks. The Koktokay Global Geopark has rare, extra-large metal deposits and mining relics, and the best-preserved relics of major earthquake fault zones locally and abroad, with these features being highly representative of the study area. Its representative scenic spots (Figure 2) include Eremu Lake, which is a typical scenic water landscape (a); Cocosuri, which is a typical scenic water, biological, and cultural landscape (b); the No. 3 Mine Pit, which is a typical scenic spot of the environmental geological landscape (c); Betula forest, which is a typical scenic biological landscape (d); and Shenzhong Canyon, which is a typical scenic geomorphic landscape (e and f). Analyses of the relationship between SEBs and EEBs and of the main problems facing the sustainable development of the Koktokay Global Geopark are not only conducive to the protection of geodiversity and biodiversity but can also promote the sustainable development of the geopark.

### 2.2. Mechanism of CCD of SEBs and EEBs

A dynamic coupling coordination relationship exists between the SEBs and EEBs of Global Geoparks; it is a complex, unbalanced, and nonlinear relationship. The two systems are linked by human activities, such as park development and environmental damage and restoration. The SEBs and EEBs not only influence and promote each other but also restrict each other, operating in contradictory unity. On the one hand, a healthy eco-environment, which includes clean water, fresh air, thick forest, a comfortable environment, rich species, etc., is a prerequisite for the sustainable development of socioeconomic activities and provides a reliable material guarantee for the sustainable growth in SEBs, thereby promoting growth in tourism revenue and resident income, and promoting the transformation of scientific research achievements. On the other hand, socioeconomic activities play a leading role in the impact of EEBs. As socioeconomic growth can provide more financial guarantees and technical support for the eco-environment and optimize the use of resources, this positive role creates better conditions for the continuous improvement in EEBs. Additionally, in the development of geoparks, tourism promotes economic development but also affects water quality, air quality, forest resources, etc., which interfere with and damage the ecosystem, leading to a series of problems, such as soil erosion, environmental pollution, biodiversity reduction, landscape damage, and so on, thereby restricting the SEBs of geoparks.

If the environment is improved, it will continue to promote the growth in SEBs. Meanwhile, the governance of and improvement in the eco-environment not only need to regularize the behavior of tourists in the process of tourism activities and implement environmental management measure but also rely on the SEBs generated by geotourism. In short, only by finding the balance between SEBs and EEBs within the acceptable threshold range of the eco-environment and maintaining their dynamic balanced development and virtuous circle can their coordinated development be realized (Figure 3).

### 2.3. Construction of Indicator System

Studies on geoparks involve many resources and disciplines such as geography, the eco-environment, and tourism. Geoparks have been established to protect geological heritage, popularize geoscience knowledge, and develop local economies. Geotourism should be reasonably developed on the premise of protecting geological heritage. Tourism drives local economic benefits, provides more financial support for the protection of geological heritage, promotes the popularization of geoscience knowledge, and, finally, achieves the original intention of protecting geological heritage. Therefore, the SEBs and EEBs produced by geoparks are special. Based on the principles of representativeness, objectivity, and comparativeness, in this study, we objectively built an indicator system that can fully reflect the comprehensive development level of SEBs and EEBs in the Koktokay Global Geopark. The system consists of two system layers: a socioeconomic benefits system (SEBs system) and an eco-environmental benefits system (EEBs system). The SEBs system consists of three primary indicators: tourism development, local economic development, and popularizing geoscience knowledge; the EEBs system consists of three primary indicators: environmental, ecological, and landscape protection.

The SEBs system reflects the contribution of geoparks to local economic development during the process of geopark construction and tourism development, and the degree to which geoparks meet social needs. We used the total tourism revenue (A_11_) to reflect the performance level of the Koktokay Global Geopark in the process of tourism development. This indicator measures the contribution of tourism development to regional economic development over time; the number of residents participating in tourism development (A_12_) is an important indicator and evaluation standard of sustainable tourism development. The local residents actively participate in geotourism development with a sense of ownership, which can enhance their sense of pride and stimulate a sustainable lifestyle. We used the Engel coefficient of residents (A_21_) to reflect the impact of the construction of the Koktokay Global Geopark and tourism development on resident income levels and living consumption, which is also an important indicator for evaluating SEBs. We used investment in planning and construction projects (A_22_) to reflect the development scale and speed of the Koktokay Global Geopark. By consulting the planning and construction projects of the park over time, we calculated the percentage of cumulative growth in total investment and cumulative growth in GDP. We selected per capita disposable income (A_23_) to reflect the impact of tourism development on the living standards and purchasing power of residents, which is an important indicator for measuring the SEBs produced by geopark construction and tourism development. When a new scientific research finding is applied to production practice, its value can be achieved. Transforming scientific research achievements into productivity (A_24_) is an important method of improving SEBs. We evaluated the contribution of scientific research achievement transformation to SEBs by calculating the ratio of scientific study findings that have been transformed to achievements to the total number of scientific research projects over the years. Activities that popularize geoscience knowledge (A_31_) effectively improve the scientific and cultural knowledge of the public, which plays an important role in promoting the optimization of industrial structure and sustainable economic development, and can be used as a flexible indicator of continuously improving SEBs.

Therefore, we selected seven secondary indicators to reflect the SEBs of the Koktokay Global Geopark: total tourism revenue (A_11_), the number of residents participating in tourism development (A_12_), the Engel coefficient of residents (A_21_), investment in planning and construction projects (A_22_), per capita disposable income (A_23_), the transformation of scientific research achievements (A_24_), and the number of popularization activities of geoscience knowledge (A_31_).

From the perspective of sustainable development, the EEB system analyzes the environmental protection and governance of geoparks in the process of geopark construction and geotourism development, and the maintenance or improvement degree of their ecological sustainable development. Water is an important part of both the environment and scenic resources in tourist destinations. However, the development of tourism usually affects water quality. Hence, we selected water cleanliness (B_11_) to measure the water quality, which we calculated by collecting the relevant monitoring data from the geopark during the peak tourism season (July to October) over time, referring to the Environmental Quality Standards for Surface Water (GB3838-2002). Air quality reflects the degree of air pollution; we selected the degree of air cleanliness (B_12_) to measure the air quality, which we calculated by obtaining the relevant monitoring data from the geopark, referring to the Ambient Air Quality Standards (GB3095-2012). The noise level (B_13_) is an important indicator for evaluating EEBs. We calculated the noise level by collecting the relevant monitoring data, referring to the Acoustic Environmental Quality Standards (GB3096-2008). Xinjiang has a shortage of water resources, and areas with relatively rich water resources are rich in vegetation, have higher air quality, and higher EEBs. When measuring the eco-environment in Xinjiang, the use of water resources is often an important measurement factor. The Koktokay Global Geopark has two main water bodies: Cocoasuri and Eremu Lakes. Hence, we used per capita water resources (B_14_) as an indicator to measure the degree of water resources use to describe the scarcity of water resources in the geopark. Species richness (B_21_) is an indicator commonly used to describe the characteristics of species diversity in biological communities and judge the stability of ecosystems. We used the Shannon–Wiener index [72] to calculate the ratio of species to individual numbers. The higher the forest coverage rate (B_22_), the richer the forest resources in the region, and the better the EEBs, which can reflect the eco-environment of the Koktokay Global Geopark. Tourism activities place pressure on the landscape of the geopark. We used landscape fragmentation (B_31_) to describe the degree to which a landscape had been disturbed by natural or human factors. The higher the fragmentation degree of the landscape, the more seriously the landscape has been damaged.

Therefore, we selected seven secondary indicators to reflect the EEBs of the Koktokay Global Geopark: water cleanliness (B_11_), degree of air cleanliness (B_12_), noise level (B_13_), per capita water resources (B_14_), species richness (B_21_), forest coverage rate (B_22_), and landscape fragmentation (B_31_) (Table 1).

### 2.4. Data Sources

We used data from the Koktokay Global Geopark from 2005 to 2018 as the research sample. We obtained most of the economy indicator data from the statistics of the Koktokay Global Geopark from 2005 to 2018, which were data relevant for planning projects and work summary reports over the period. The other data regarding environment indicators, such as water cleanliness, degree of air cleanliness, and noise level, were obtained from the relevant monitoring data of the geopark from July to October during the tourism season each year. We conducted the calculations to national standards such as the Environmental Quality Standards for Surface Water (GB3838-2002), Ambient Air Quality Standards (GB3095-2012), and Acoustic Environmental Quality Standards (GB3096-2008). We obtained the species richness indicator by accessing the master plan of the Koktokay eco-tourism demonstration area (2017–2026) and the recent comprehensive investigation reports of the geopark. We supplemented missing data using the tourism and ecological civilization data from the Koktokay Global Geopark.

### 2.5. Method

#### 2.5.1. Index Standardization and Weight Determination

(1)Data standardization

Due to the differences in the attributes and dimensions among the evaluation indicators, we used the range method for the dimensionless processing of all data to ensure that they were measurable. We divided the data into positive and negative indicators. Positive indicators had a positive impact on the system, and vice versa. Then, we chose Formulas (1) and (2) to standardize the data [73]:(1)$$\mathrm{Positive}\;\mathrm{indicator}:\;X_{ij}=\frac{x_{ij}-\min x_{ij}}{\max x_{ij}-\min x_{ij}}$$
(2)$$\mathrm{Negative}\;\mathrm{indicator}:\;X_{ij}=\frac{\max x_{ij}-x_{ij}}{\max x_{ij}-\min x_{ij}}$$
where $X_{ij}$ is the actual value of each indicator; $\max x_{ij}$ and $\min x_{ij}$ are the maximum and minimum values of indicator j for the ith year, respectively.

(2)Weight calculation

We used the entropy method to objectively determine the weight of each indicator, which is based on the amount of information provided by each factor. The larger the amount of information, the smaller the uncertainty. This is a method of judging the degree of order and its effectiveness according to the amount of information obtained. The calculation steps are as follows [74]:

Step 1: Calculate the proportion of indicators:(3)$$p_{ij}=\frac{X_{ij}}{\sum_{i=1}^{n}X_{ij}}$$

Step 2: Calculate the information entropy of indicators:(4)$$E_{j}=-\ln{\left(m\right)}^{-1}\sum_{i=1}^{m}p_{ij}\ln\left(p_{ij}\right)$$

Step 3: Calculate the weight of indicators:(5)$$W_{j}=\left(1-E_{j}\right)/\left(k-\sum_{i=1}^{m}E_{j}\right)$$
where $p_{ij}$ represents the proportion of indicator j in the ith year, m is the number of indicators, n is the year, k is the number of indicators in the subsystem, and $E_{j}$ is the information entropy of each indicator. The higher the value of $E_{j}$, the larger the amount of information it carries and the stronger the uncertainty of the system. We calculated the weights of the SEBs $w_{Aj}$ and EEBs $w_{Bj}$ of the Koktokay Global Geopark; the results are presented in Table 2 and Table 3.

#### 2.5.2. Analysis of the CCD Model

(1)Comprehensive evaluation index

We used the standardized values of the two systems, which we obtained using the range method, and their respective weights to calculate the comprehensive evaluation index by using the linear weighting method. The formula is as follows:(6)$$F(x)=\sum_{i=1}^{m}w_{Ai}X_{Ai}$$
(7)$$F(y)=\sum_{i=1}^{m}w_{Bi}X_{Bi}$$
where F(x) and F(y) represent the comprehensive evaluation indexes of the SEBs and EEBs systems, respectively, of the Koktokay Global Geopark; $X_{Ai}$ and $X_{Bi}$ are the standardized values of the SEBs and EEBs systems of the geopark, respectively; $w_{Ai}$ and $w_{Bi}$ are the weight values calculated by the two systems, respectively.

(2)CCD model

To further evaluate the coupling degree of the two systems, we introduced the coupling evaluation model in physics, which we used to calculate the coupling degree between the two systems to reflect their interactions and judge whether development was orderly [75]:(8)$$C=\sqrt{\frac{F(x)\times F(y)}{{\left[F(x)+F(y)\right]}^{2}}}$$
where C refers to the coupling degree of the SEBs and EEBs systems of the Koktokay Global Geopark, whose value is within the range [0, 1]. When the value of *C* is larger, the degree of coupling between two systems is higher. When the value is smaller, the two systems are less coupled, which means that they may be in a state of disorder or imbalance.

The results of coupling degree analysis only describe the degree of interaction among the systems. To fully reflect the CCD of the SEBs and EEBs systems of the geopark, we needed to further analyze the coordination degree of the coupling relationship between the SEBs and EEBs systems of the geopark with the help of the CCD model, so we could judge whether the two systems are harmoniously developing [50]. The formula is as follows:(9)$$D=\sqrt{C\cdot T}$$
(10)T=rF(x)+qF(y)
where D denotes the CCD of the SEB and EEB systems of the geopark. When the value of *D* is larger, the coordinated development degree of the system is higher, and vice versa. T is the comprehensive coordination index, which reflects the comprehensive development status of the SEBs and EEBs systems of the geopark; r and q are the weights of the SEB and EEB systems, respectively. To balance the development of the two systems, we set their value to 0.5 based on a former study [76].

Referring to several previous studies [57,73,74], in this study, we divided the CCD of the SEBs and EEBs systems of the geoparks into 2 categories, 10 subclasses, and 30 types. The classification of the CCD is shown in Table 4. D was the CCD (0≤D≤1), which we divided into two categories: disorder and coordination. Disorder means imbalance, which referred to the imbalance between the SEBs and EEBs system in this study. Coordination is a benign correlation between two or more subsystems, which guarantees the healthy development of the system. In this study, coordination denoted that the SEBs and EEBs systems were in a benign interactive relationship. According to the degree of harmony between the two systems, we then further divided the CCD into 10 subclasses. In addition, we divided the development types into three kinds: EEBs-lagging, SEBs-lagging, and SEBs and EEBs synchronized, according to the different values of *F*(*x*) and *F*(*y*). When *F*(*x*) > *F*(*y*), the development was categorized as the EEBs-lagging type, indicating that the EEBs system had fallen behind the SEBs system, with low-level EEBs and high-level SEBs. When *F(x)* < *F*(*y*), the development was categorized as the SEBs-lagging type, indicating that the SEBs system had fallen behind the EEBs system, with low-level SEBs and high-level EEBs. When *F*(*x*) = *F*(*y*), the development was categorized as the SEBs and EEBs synchronized type, indicating that the development speed and direction of the two systems were consistent, harmoniously achieving benefits.

## 3. Results

### 3.1. Analysis of the Development Level of the Socioeconomic Benefits (SEBs) and Eco-Environmental Benefits (EEBs) Systems

#### 3.1.1. Analysis of Comprehensive the Development Level of the SEBs System

The comprehensive development level of the SEBs of the Koktokay Global Geopark showed a fluctuating upward trend during the study period. The comprehensive evaluation index of SEBs increased from 0.1989 in 2005 to 0.6645 in 2018, an increase of nearly 2.34 times. According to the change in the value of the comprehensive evaluation index of SEBs, we divided the SEBs into main four stages, as shown in Figure 4.

First, 2005–2008 was a period of rapid development, during which the comprehensive evaluation index of SEBs increased from 0.1989 in 2005 to 0.4590 in 2008, an increase of nearly 1.31 times. Since the Koktokay Global Geopark was approved as a National Geopark in 2005, the local government has continuously improved the infrastructure in the geopark. At the same time, the government has paid attention to protecting geological heritage resources and designed new tourist routes, in accordance with the brand management strategy. Adhering to the principle of “protection first and development second”, the harmonious unity of infrastructure construction and natural environment protection must be achieved. During this period, the number of tourists rapidly increased, and tourism revenue increased from CNY 32.0921 million to CNY 40.513 million. Therefore, the socioeconomic development of the Koktokay Global Geopark has increasingly improved, and the comprehensive development level of the SEBs system of the geopark has continuously risen during the four-year period.

Second, 2009–2012 was the first phase of adjustment, during which the comprehensive evaluation index of SEBs increased slowly from 0.4170 in 2009 to 0.5698 in 2012, a small overall increase. Affected by the 2009 event in Urumqi, Xinjiang’s tourism economy significantly decreased. Tourism development in the Altay region and the SEBs of the Koktokay Global Geopark were also affected. Economic development has gradually recovered since 2010. The government has continuously increased its investment in tourism development planning and construction projects, focusing on the upgrading of scenic spots and performing Global Geoparks work. After being successfully selected as a national 4A scenic spot in 2009, the Koktokay Global Geopark was upgraded to a national 5A scenic spot in 2012. The rapid development of tourism has increased the number of local residents participating in tourism and gradually improved their living standards. The improvement in regional visibility has also attracted many visiting experts and scholars, which has not only promoted the transformation of scientific research achievements of the geopark but also provided favorable conditions for popularizing geoscience knowledge. In the 3 years from 2010 to 2012, the SEBs of the Koktokay Global Geopark rose from 0.4906 to 0.5698.

Third, 2013–2014 was the second phase of adjustment. The comprehensive evaluation index of SEBs declined again in 2013 and rebounded slightly to 0.5460 in 2014. The reason for this was the events endangering social security in southern Xinjiang in 2013, which seriously affected the socioeconomic development of Xinjiang.

Fourth, 2015–2018 was a recovery period, during which the comprehensive evaluation index of the SEBs gradually increased from 0.6074 in 2015 to 0.6645 in 2018. With the improvement in the social situation, the economic tourism connections among different regions have gradually strengthened. Owing to the strong support of the government, and the rich scientific connotations and unique landscape of the geopark, the geopark was successfully approved as a Global Geopark in 2015, which increased the number of people realizing the value of the Koktokay Global Geopark, attracted more tourists, and enhanced the pride of local residents. Furthermore, more relevant stakeholders became involved in geopark activities. Related economic entities, such as catering, transportation, retail, and other service industries driven by tourism, also benefited. In these four years, the investment in planning, construction, and development of the Koktokay Global Geopark substantially increased; the service level gradually improved; and more geoproducts and activities were provided for the popularization of geoscience knowledge, such as geological museums, youth research activities, geoscience summer camps, red gene theme education, etc. In addition, during the promotion of the socioeconomic development of the geopark, local residents participated in the construction of the geopark, which was conducive to solving the problem of employment and transferring surplus labor force in local communities. The Koktokay Global Geopark has gradually shifted to promoting a steady rise in the comprehensive SEBs level from three aspects: tourism, local economic development, and the popularization of geoscience knowledge.

#### 3.1.2. Analysis of the Comprehensive Development Level of the EEBs System

The comprehensive development level of the EEBs system of the Koktokay Global Geopark generally remained stable but fluctuated slightly from 2005 to 2018, showing a slightly U-shaped evolution. According to the changes in its comprehensive evaluation index, we divided the EEBs system development into four stages, as shown in Figure 3.

The first stage was a stable period from 2005 to 2008, during which the comprehensive evaluation index of EEBs remained in the range of 0.3081–0.3240. Tourism at the Koktokay Global Geopark was still in its infancy in 2015, and planning and construction projects were gradually beginning to thrive; consequently, the construction of infrastructure began to put pressure on the eco-environment. In 2007, the comprehensive development level of the EEBs of the geopark slightly decreased, but overall, the protection and management of the eco-environment were maintained at a relatively stable level during these four years.

The second stage was the decline period, during which the comprehensive evaluation index of EEBs decreased from 0.3023 in 2009 to 0.2119 in 2012, showing a downward trend. The geopark strengthened infrastructure construction to promote the development of tourism. With the growth in the popularity of the geopark, the number of tourists increased each year, consequently increasing the pressure on the eco-environment. In 2009, the geopark received more than 300,000 tourist visits. The continuous interference from human activities and the implementation of geopark planning and construction projects damaged forest coverage and biodiversity, increasing the fragmentation of the eco-environment, and thereby resulting in a gradual decline in EEBs. With the increase in tourists, the eco-environment of the geopark faced more serious challenges. The economic growth produced by increases in tourism was used for further infrastructure reconstruction in the geopark, further damaging the geological heritage and eco-environment. In addition, many tourists’ inappropriate behaviors during their leisure activities led to environmental pollution of the geopark. From 2009 to 2012, the eco-environment deteriorated, and the comprehensive evaluation index of the EEBs fell to its lowest point in 2012.

The third stage, from 2013 to 2014, was the recovery period during which the comprehensive evaluation index of EEBs increased and recovered to 0.3191 in 2014. With the start of the work of applying for Global Geopark status, the geopark’s managers strengthened their efforts to protect the eco-environment, such as through air pollution prevention and control, lake eco-environment and wetland protection, and restoration. In addition, they strengthened the evaluations and assessments of the geopark, improved public sanitation facilities, and strengthened the management of tourists, thereby gradually improving its EEBs.

The fourth stage was the steady development period from 2014 to 2018, during which the comprehensive evaluation index of EEBs remained within a higher range of 0.3045–0.3691. The approval of the Koktokay Geopark as a Global Geopark in 2015 led to the continuous increase in the comprehensive evaluation index of the EEBs, and the tourism development of the geopark entered a new stage. Under the ongoing human activities, the EEBs fluctuated within a small range, and environmental, ecological, and landscape protection gradually improved. Adhering to the goal of maintaining a healthy eco-environment set forth by UNESCO, the comprehensive development level of the EEBs of the Koktokay Global Geopark gradually improved and stabilized.

A further analysis showed that the SEBs system of the Koktokay Global Geopark F(x), although fluctuating, gradually improved, as did the EEBs system F(y), again after a slow decrease in volatility. The comprehensive development level showed evolutionary characteristics, which we divided into four stages: 2005–2008, 2009–2012, 2013–2014, and 2015–2018. The SEBs system experienced improvements from fast to slow: a rapid development period from 2005 to 2008, when the EEBs system was relatively stable. From 2007 to 2008, the comprehensive evaluation index of SEBs was substantially higher than that of EEBs, showing a certain gap in their development.

From the development direction perspective, from 2009 to 2012, while the comprehensive evaluation index of SEBs increased, the EEBs trended downward. In addition, in the following years, the SEBs increased and the EEBs decreased, which reflected the situation in reality, given the sensitivity of the eco-environment of the Koktokay Global Geopark to human interference.

From the perspective of changes, the SEBs of the Koktokay Global Geopark changed more quickly. Although the comprehensive evaluation index of the EEBs substantially decreased from 2008 to 2012, its change in speed was slower than that of the comprehensive evaluation index of the SEBs, which showed that the local department had implemented effective management of the geopark. In the 14-year period, when the SEBs rapidly grew, although the EEBs fluctuated, they were still able to be maintained at a relatively stable level.

### 3.2. CCD Analysis of SEBs and EEBs Systems

We calculated the comprehensive evaluation indexes of the SEBs and EEBs systems of the Koktokay Global Geopark from 2005 to 2018 F(x) and F(y), the coupling degree C, and the comprehensive coordination index T using the CCD model. From this, we obtained the coupling degree D. The calculation results are shown in Table 5.

As can be seen from Figure 5, the CCD between the SEBs and EEBs systems of the Koktokay Global Geopark increased from 0.3814 in 2005 to 0.5015 in 2018, indicating that the relationship between the two systems was gradually becoming coordinated. Specifically, in the interaction process between the SEBs and EEBs systems of the geopark from 2005 to 2007, T gradually increased, but C was stable at first and then decreased, thereby affecting the fluctuations in D. From 2007 to 2009, the motion trajectories of F(x) and F(y) gradually converged, the development trajectories of D and T also gradually converged, and C showed a steady upward trend. From 2010 to 2012, with the gradual separation of the motion trajectories of the two systems, the motion trajectories of C showed a gradual downward trend, and the motion trajectory of T did not fluctuate, so D also showed a downward trend. From 2013 to 2016, as the comprehensive development index of the two systems gradually approached and separated, the values of C, T, and D tended to rise or fall at the same time. From 2017 to 2018, F(x) and F(y) tended to be stable or rose, and the values of C, T, and D showed a trend of gradually approaching each other.

Overall, the development trends in C, T, and D in the 14-year period were converging, indicating that the CCD of the SEBs and EEBs systems is becoming increasingly strong, with improved development occurring. Except for 2009, the curve movement trends in C and D in other years were basically consistent, which also showed that the CCD of the SEBs and EEBs systems were strongly correlated with their development level.

### 3.3. Type Analysis of CCD of SEBs and EEBs Systems

After calculating F(x), F(y), C, T, and D through the coupling coordination model, we classified the types of CCD of the SEBs and EEBs systems of the Koktokay Global Geopark (Table 5). As shown in Figure 6, the type of CCD of the SEBs and EEBs systems of the Koktokay Global Geopark were classified as mildly disordered from 2005 to 2007. Although the degree of disorder improved in 2006, the overall situation has not changed. While the SEBs rapidly rose, the EEBs were always stable; for that reason, the comprehensive evaluation index of the EEBs system has lagged. From 2008 to 2015, the type of CCD transitioned to the near disorder stage, but the overall trend was still improving, finally entering the barely coordinated stage in 2016. The SEBs system, at this stage, was always ahead of the EEBs system, but the comprehensive evaluation index of the EEBs system, overall, showed a continuous upward trend. So, when the coordination degree of the SEBs and EEBs systems of the Koktokay Global Geopark entered the barely coordinated stage, the SEBs system assumed the dominant role, and its interactions with the EEBs system also improved, showing a trend of gradual convergence. In general, the type of CCD of the SEBs and EEBs systems of the Koktokay Global Geopark has transitioned, showing that the CCD between the SEBs and EEBs systems has been developing in an increasingly coordinated direction.

## 4. Discussion

By constructing an indicator system for evaluating the SEBs and EEBs of geoparks, we discussed the characteristics of the variation in and the coupling coordination relationship between the SEBs and EEBs of the Koktokay Global Geopark from 2005 to 2018. From the results, we found that from 2005 to 2018, the SEBs of the geopark showed a fluctuating upward trend, and experienced three stages (rapid development, adjustment, and recovery), indicating that the government’s investment in tourism development, residents’ participation, and the social situation strongly impact the SEBs of geoparks. Therefore, a more reasonable geopark management system must be established according to the actual situations in various countries; a favorable policy environment must be created for the development of geoparks; and the interests of local residents, developers, the government, and other stakeholders must be coordinated through a scientific management model. The EEBs remained stable on the whole but slightly fluctuated, and we found it underwent four stages of steady–decline–recovery–steady development. The decline occurred in 2009–2012 due to the deterioration of the eco-environment caused by tourism development. Later, with the application for UNESCO Global Geoparks status and the attention paid by geopark managers to the eco-environment, the EEBs gradually improved. This showed that the ecological protection measures and regulations implemented by local governments during the application for the status and construction of Global Geoparks play an active role in the eco-environment protection of geoparks, which is a finding similar to that previously reported [77]. Moreover, the CCD of the SEBs and EEBs systems generally showed a fluctuating upward state. The type of CCD transitioned from mild disorder to the nearly disordered stage, and then entered the barely coordinated stage, gradually showing a trend in increased coupling development, which is consistent with the conclusions reported by Yi et al. [70]. This shows that if the eco-environment is properly protected, the SEBs and EEBs systems change from the disordered to coordinated stages, which has become the main feature of most geoparks during their development process in recent years. Our findings showed that the economic activities associated with the development of geoparks not only generate jobs and income but also raise public awareness of the sustainable management of precious geological heritage, which is consistent with reported findings [17]. However, to achieve sustainable development, landscape and ecological security must be prioritized in the context of geopark tourism and economic development [77].

The Koktokay Global Geopark is similar to other UNESCO Global Geoparks [78,79]. These parks are rich in geological resources and can improve the living conditions of local residents and the eco-environment by encouraging local communities to participate in geopark activities, including cultural and recreational activities. In addition, the geoparks can provide accessible knowledge regarding geological heritage to local residents and visitors; they can thus be used as educational tools. Therefore, the Global Geoparks have many SEBs and EEBs. The Koktokay Global Geopark has its own unique qualities: It is located in China, the country with the largest number of Global Geoparks. It was the first Global Geopark in China, having typical mineral deposits and mine sites as its main landscape, and contains rare global mining relics, 86 kinds of known minerals, more than 140 kinds of useful minerals, and a variety of gemstones. The Koktokay Global Geopark is a comprehensive natural park integrating scientific research, education, and tourism with geodiversity, cultural diversity, and biodiversity. The practices used for the development of geoparks show that the protection of geological heritage and the development of geotourism complement each other. Cities [42], nature reserves [80], and wetland parks [81] have similar characteristics. Only by coordinating the relationship between socioeconomic development and the eco-environment can regional sustainable development be achieved. Not only does the Koktokay Global Geopark need to combine resource development with ecological environment protection but so should other Global Geoparks to coordinate SEBs and EEBs. Therefore, this study provides a reference for other UNESCO Global Geoparks.

Our main purpose in this study was to explore the interactions and coupling relationship between the SEBs and EEBs of geoparks. Previously, researchers mostly studied cases to unilaterally analyze the socioeconomic impact [17] or eco-environment status [77] of a Global Geopark, but studies on the relationship between them by building a more sophisticated index system are lacking. Based on relevant research results regarding the coordinated development of the regional economy and environment [82] and of ecological and economic systems [83], and the impact of tourism activities on the eco-environment [84], combined with the characteristics of geoparks, in this study, we constructed an indicator system of the SEBs and EEBs of geoparks, and applied it to the Koktokay Global Geopark to test its scientificity. Our method provides a useful reference for the study of the dynamic relationship between the SEBs and EEBs of other similar geoparks or tourist destinations. In addition, we designed a model for evaluating the SEBs and EEBs of geoparks, and we formulated the division of types and evaluation criteria of coupling coordination development, which allowed a more scientific and accurate judgement of the development state in each period. The evolution of the curve of the comprehensive evaluation index of the SEBs and EEBs, and of the CCD demonstrated the development of geoparks and reflected the problems being faced regarding their protection and development. Considering the dynamics, complexity, openness, and imbalance of the SEBs and EEBs systems of geoparks, the model of evaluating CCD needs to be further verified and improved, and the evaluation criteria need to be further specified and studied.

## 5. Conclusions

The past few decades have witnessed extensive geopark developments, with geopark managers needing to protect the eco-environment while ensuring economic development. Therefore, the coordinated development of socioeconomic conditions and the eco-environment is crucial for the sustainable development of geoparks. Based on coupling coordination degree (CCD) theory, we used the Koktokay Global Geopark as a case study area to establish an indicator system for evaluating the CCD between this geopark’s socioeconomic benefits (SEBs) and eco-environmental benefits (EEBs), and to discuss the variations in the characteristics and the CCD types of the two systems. Our main conclusions are as follows:
(1)The Koktokay Global Geopark’s SEBs showed a fluctuating upward trend during the study period; this process trended from fast to slow. The change in the speed of the comprehensive evaluation level of SEBs was faster than that of the whole; specifically, 2005–2008 was a period of rapid development.(2)The Koktokay Global Geopark’s EEBs remained stable but slightly fluctuated during the study period, showing a slightly U-shaped evolution trend. In particular, it declined from 2009 to 2012, during which the development of geotourism not only led to economic growth but also to the deterioration of the eco-environment, causing issues such as water and air pollution and decreases in biodiversity. The comprehensive development index of the EEBs of the Koktokay Global Geopark fell to its lowest point in 2012, then gradually improved and stabilized as the eco-environmental protection of the geopark strengthened.(3)In terms of changes in the CCD between the SEBs and EEBs, overall, we observed an upward fluctuation: the value increased from 0.3814 in 2005 to 0.5015 in 2018, indicating that the two systems were developing in an increasingly coordinated direction.

With the rapid development of the social economy and geotourism, some problems are facing geopark development, such as the destruction of geological heritage, the lack of innovative geoproducts, the decline in environmental capacity, and so on. We therefore provide the following thoughts and suggestions.

First, in the development of geoparks, we should not ignore their environmentally friendly development in order to gain immediate benefits. We advise pursuing scientific and rational development to protect geological heritage and the ecological environment. Specifically, the multiple relevant stakeholders, including geopark management agencies, local communities, government authorities, local businesses, and academic and research institutions, need to actively participate and maintain good relationships in the geopark development process. They need to not only protect geological heritage but also pay attention to local water and air quality, forest resources, and biodiversity.

Second, geological heritage needs to be protected in the process of development, and scientific planning is required for processes such as dividing protected areas and tourist areas, which contribute to the socioeconomic and sustainability of a region.

Third, we recommend building popular science bases or holding geoeducation activities, which integrate science education with tourism, popularize geoscience knowledge among the public, and demonstrate the scientific value of tourism resources. In addition, geoeducation activities provide educational opportunities to enhance the public’s awareness of precious geological resources and improve the public’s enthusiasm to participate in science and public environmental education.

Fourth, we suggest that academic and research institutions cooperate to continuously develop new methods to promote and understand geological heritage, that new geological tourism products should be developed, and that scientific research achievements should be transformed into SEBs.

The construction of geoparks also needs to consider local economic and natural conditions, take advantage of local characteristics, and drive the development of tourism, thereby improving the local economic development level and residents’ living standards, which will enhance residents’ awareness of protecting geological heritage and finally achieve sustainable and coordinated development between socioeconomic conditions and the eco-environment. Considering the three functions of geoparks—the protection of geological heritage, popularizing geoscience knowledge, and the development of the local economy—we constructed an indicator system of SEBs and EEBs in the Koktokay Global Geopark and calculated their CCD, providing a reference for the sustainable development of other similar geoparks.

## Figures and Tables

**Figure 1 ijerph-19-08498-f001:**
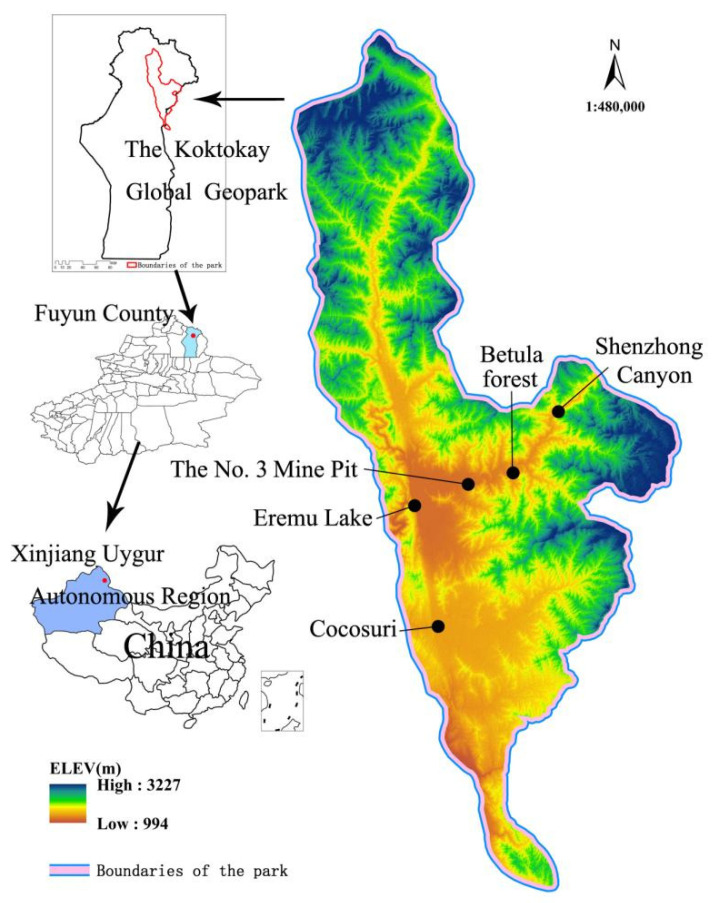
Map of Koktokay Global Geopark showing geological and heritage resources and elevation changes, indicating its location within northwestern China.

**Figure 2 ijerph-19-08498-f002:**
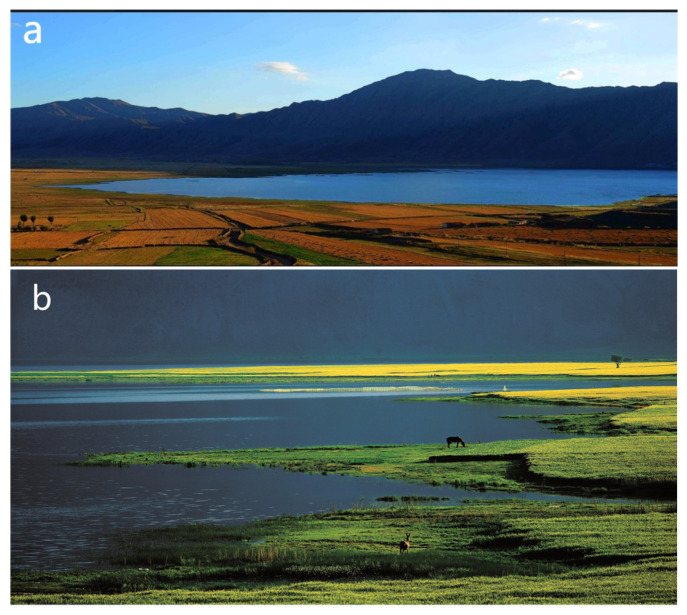
The representative scenic spots of the Koktokay Global Geopark. (**a**) Eremu Lake; (**b**) Cocosuri; (**c**) the No. 3 Mine Pit; (**d**) Betula forest; (**e**,**f**) Shenzhong Canyon.

**Figure 3 ijerph-19-08498-f003:**
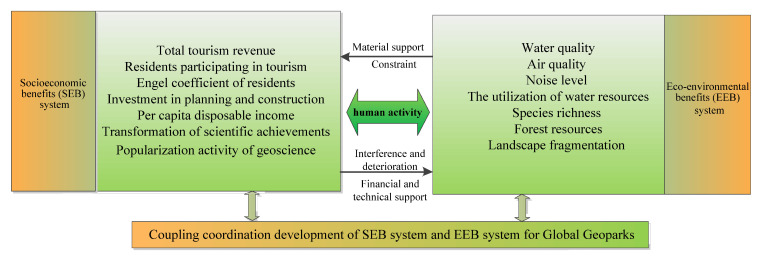
Complex system of SEBs and EEBs of Global Geoparks.

**Figure 4 ijerph-19-08498-f004:**
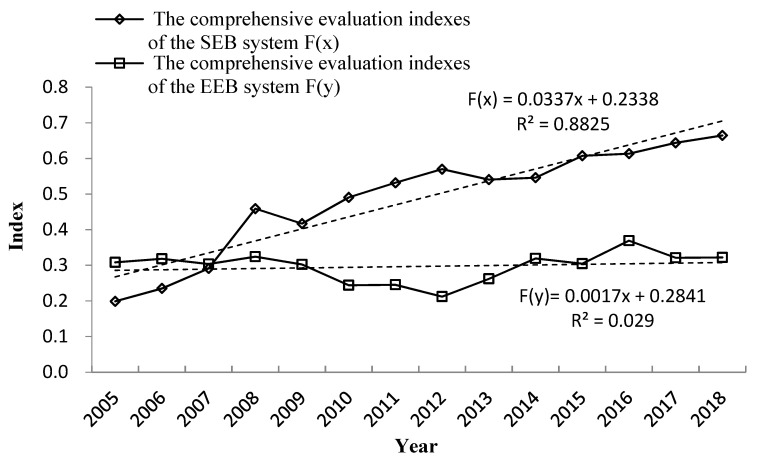
Evolution of comprehensive evaluation index of SEBs and EEBs of Koktokay Global Geopark from 2005 to 2018.

**Figure 5 ijerph-19-08498-f005:**
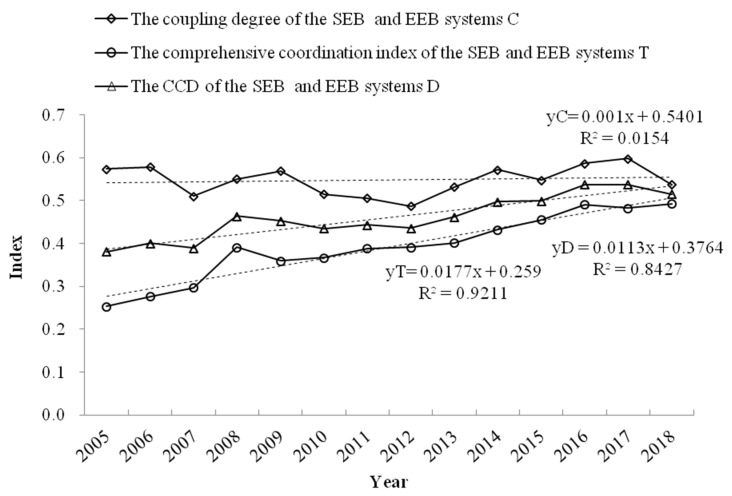
Evolving curve of the CCD of the SEBs and EEBs systems for the Koktokay Global Geopark from 2005 to 2018.

**Figure 6 ijerph-19-08498-f006:**
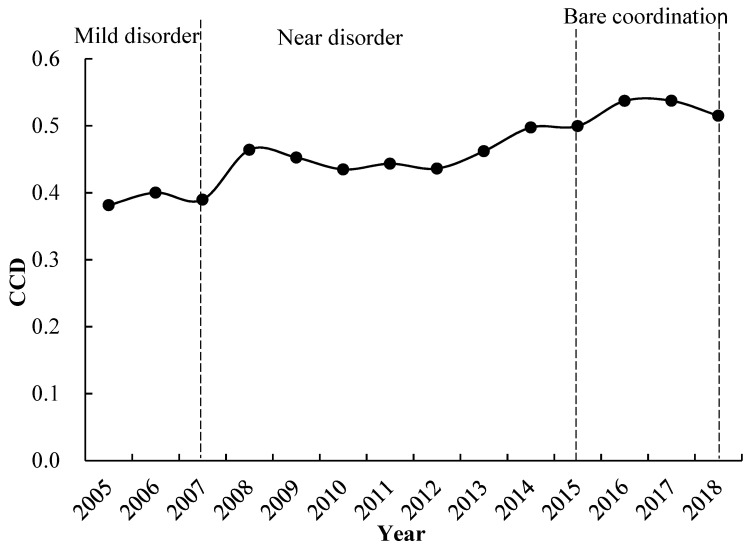
CCD type between the SEBs and EEBs systems of Koktokay Global Geopark.

**Table 1 ijerph-19-08498-t001:** The indicator system of SEBs and EEBs.

Subsystem	First-Level Indicator	Second-Level Indicator	Type
SEB system (A)	Tourism development A_1_	Total tourism revenue A_11_	+
		Number of residents participating in tourism development A_12_	+
	Local economic development A_2_	Engel coefficient of residents A_21_	−
		Investment in planning and construction projects A_22_	+
		Per capita disposable income A_23_	+
		Transformation of scientific research achievements A_24_	+
	Popularizing geoscience knowledge A_3_	Number of popularization activities of geoscience knowledge A_31_	+
EEB system (B)	Environmental protection B_1_	Water cleanliness B_11_	+
		Degree of air cleanliness B_12_	+
		Noise level B13	−
		Per capita water resources B_14_	+
	Ecological protection B_2_	Species richness B_21_	+
		Forest coverage rate B_22_	+
	Landscape protection B_3_	Landscape fragmentation B_31_	−

Note: “+” indicates a positive indicator; “−” indicates a negative indicator.

**Table 2 ijerph-19-08498-t002:** Information entropy and weight of each indicator of SEBs.

Year	A_1_ (0.0316)		A_2_ (0.0434)				A_3_ (0.0122)
	A_11_	A_12_	A_21_	A_22_	A_23_	A_24_	
2005	0.0229	0.0000	0.0362	0.0323	0.0031	0.0000	0.0000
2006	0.0163	0.0000	0.0371	0.0203	0.0029	0.0060	0.0000
2007	0.0248	0.0000	0.0375	0.0236	0.0033	0.0109	0.0000
2008	0.0202	0.0000	0.0283	0.0184	0.0035	0.0308	0.0003
2009	0.0261	0.0031	0.0317	0.0205	0.0031	0.1327	0.0005
2010	0.0183	0.0000	0.0238	0.0176	0.0041	0.0217	0.0004
2011	0.0287	0.0104	0.0204	0.0146	0.0051	0.1149	0.0026
2012	0.0161	0.1275	0.0316	0.0155	0.0059	0.1325	0.0135
2013	0.0218	0.0036	0.0274	0.0304	0.0051	0.1475	0.0108
2014	0.0676	0.0310	0.0251	0.0185	0.0043	0.1155	0.0232
2015	0.0318	0.0290	0.0284	0.0244	0.0052	0.2283	0.0183
2016	0.0612	0.0349	0.0241	0.0296	0.0073	0.2010	0.0308
2017	0.0915	0.0361	0.0221	0.0282	0.0064	0.2134	0.0347
2018	0.1242	0.0384	0.0236	0.1104	0.0069	0.2078	0.0358
*E_j_*	0.2937	0.1614	0.2042	0.2078	0.0340	0.8033	0.0878
*w_Aj_*	0.1312	0.1558	0.1478	0.1472	0.1795	0.0365	0.1695

**Table 3 ijerph-19-08498-t003:** Information entropy and weight of each indicator of EEBs.

Year	B_1_ (0.0450)				B_2_ (0.0057)		B_3_ (0.0273)
	B_11_	B_12_	B_13_	B_14_	B_21_	B_22_	
2005	0.0608	0.0039	0.0302	0.0202	0.0042	0.0073	0.0260
2006	0.0614	0.0042	0.0308	0.0192	0.0042	0.0080	0.0262
2007	0.0687	0.0055	0.0316	0.0247	0.0039	0.0076	0.0285
2008	0.0684	0.0054	0.0296	0.0166	0.0044	0.0070	0.0281
2009	0.0501	0.0068	0.0304	0.0239	0.0043	0.0070	0.0270
2010	0.0473	0.0073	0.0314	0.0263	0.0037	0.0069	0.0278
2011	0.0515	0.0113	0.0321	0.0460	0.0037	0.0067	0.0261
2012	0.0460	0.0253	0.0308	0.0348	0.0038	0.0070	0.0296
2013	0.0629	0.0476	0.3029	0.0616	0.0043	0.0068	0.0280
2014	0.0490	0.0236	0.0285	0.1105	0.0042	0.0068	0.0266
2015	0.0478	0.0192	0.0307	0.1134	0.0039	0.0070	0.0278
2016	0.0419	0.0201	0.0314	0.0701	0.0044	0.0070	0.0262
2017	0.0483	0.0337	0.0297	0.1362	0.0044	0.0074	0.0273
2018	0.0507	0.0413	0.0310	0.1078	0.0043	0.0081	0.0264
*E_j_*	0.3880	0.1312	0.3603	0.4170	0.0297	0.0517	0.1961
*w_Bj_*	0.1137	0.1614	0.1188	0.1083	0.1802	0.1762	0.1493

**Table 4 ijerph-19-08498-t004:** Classification of CCD of SEBs and EEBs systems of geoparks.

Category	CCD	Subclass	Type	Characteristic
Disorder	0.00–0.09	Extreme disorder	*F*(*x*) > *F*(*y*)	Extreme disorder with EEBs lagging
			*F*(*x*) = *F*(*y*)	Extreme disorder with SEBs and EEBs synchronized
			*F*(*x*) < *F*(*y*)	Extreme disorder with SEBs lagging
	0.1–0.19	Serious disorder	*F*(*x*) > *F*(*y*)	Serious disorder with EEBs lagging
			*F*(*x*) = *F*(*y*)	Serious disorder with SEBs and EEBs synchronized
			*F*(*x*) < *F*(*y*)	Serious disorder with SEBs lagging
	0.2–0.29	Moderate disorder	*F*(*x*) > *F*(*y*)	Moderate disorder with EEBs lagging
			*F*(*x*) = *F*(*y*)	Moderate disorder with SEBs and EEBs synchronized
			*F*(*x*) < *F*(*y*)	Moderate disorder with SEBs lagging
	0.3–0.39	Mild disorder	*F*(*x*) > *F*(*y*)	Mild disorder with EEBs lagging
			*F*(*x*) = *F*(*y*)	Mild disorder with SEBs and EEBs synchronized
			*F*(*x*) < *F*(*y*)	Mild disorder with SEBs lagging
	0.4–0.49	Near disorder	*F*(*x*) > *F*(*y*)	Near disorder with EEBs lagging
			*F*(*x*) = *F*(*y*)	Near disorder with SEBs and EEBs synchronized
			*F*(*x*) < *F*(*y*)	Near disorder with SEBs lagging
Coordination	0.5–0.59	Bare coordination	*F*(*x*) > *F*(*y*)	Bare coordination with EEBs lagging
			*F*(*x*) = *F*(*y*)	Bare coordination with SEBs and EEBs synchronized
			*F*(*x*) < *F*(*y*)	Bare coordination with SEBs lagging
	0.6–0.69	Primary coordination	*F*(*x*) > *F*(*y*)	Primary coordination with EEBs lagging
			*F*(*x*) = *F*(*y*)	Primary coordination with SEBs and EEBs synchronized
			*F*(*x*) < *F*(*y*)	Primary coordination with SEBs lagging
	0.7–0.79	Intermediate coordination	*F*(*x*) > *F*(*y*)	Intermediate coordination with EEBs lagging
			*F*(*x*) = *F*(*y*)	Intermediate coordination with SEBs and EEBs synchronized
			*F*(*x*) < *F*(*y*)	Intermediate coordination with SEBs lagging
	0.8–0.89	Good coordination	*F*(*x*) > *F*(*y*)	Good coordination with EEBs lagging
			*F*(*x*) = *F*(*y*)	Good coordination with SEBs and EEBs synchronized
			*F*(*x*) < *F*(*y*)	Good coordination with SEBs lagging
	0.9–1.00	High-quality coordination	*F*(*x*) > *F*(*y*)	High-quality coordination with EEBs lagging
			*F*(*x*) = *F*(*y*)	High-quality coordination with SEBs and EEBs synchronized
			*F*(*x*) < *F*(*y*)	High-quality coordination with SEBs lagging

**Table 5 ijerph-19-08498-t005:** CCD of the SEBs and EEBs systems for the Koktokay Global Geopark from 2005 to 2018.

Year	*F*(*x*)	*F*(*y*)	*C*	*T*	*D*	Type	Characteristic
2005	0.1989	0.3081	0.5739	0.2535	0.3814	*F*(*x*) < *F*(*y*)	Mild disorder with SEBs lagging
2006	0.2350	0.3182	0.5790	0.2766	0.4002	*F*(*x*) < *F*(*y*)	Near disorder with SEBs lagging
2007	0.2912	0.3036	0.5107	0.2974	0.3897	*F*(*x*) < *F*(*y*)	Mild disorder with SEBs lagging
2008	0.4590	0.3240	0.5504	0.3915	0.4642	*F*(*x*) > *F*(*y*)	Near disorder with EEBs lagging
2009	0.4170	0.3023	0.5696	0.3596	0.4526	*F*(*x*) > *F*(*y*)	Near disorder with EEBs lagging
2010	0.4906	0.2438	0.5151	0.3672	0.4349	*F*(*x*) > *F*(*y*)	Near disorder with EEBs lagging
2011	0.5317	0.2453	0.5061	0.3885	0.4434	*F*(*x*) > *F*(*y*)	Near disorder with EEBs lagging
2012	0.5698	0.2119	0.4868	0.3909	0.4362	*F*(*x*) > *F*(*y*)	Near disorder with EEBs lagging
2013	0.5406	0.2618	0.5323	0.4012	0.4621	*F*(*x*) > *F*(*y*)	Near disorder with EEBs lagging
2014	0.5460	0.3191	0.5722	0.4326	0.4975	*F*(*x*) > *F*(*y*)	Near disorder with EEBs lagging
2015	0.6074	0.3045	0.5474	0.4559	0.4996	*F*(*x*) > *F*(*y*)	Near disorder with EEBs lagging
2016	0.6134	0.3691	0.5874	0.4913	0.5372	*F*(*x*) > *F*(*y*)	Bare coordination with EEBs lagging
2017	0.6439	0.3210	0.5986	0.4824	0.5374	*F*(*x*) > *F*(*y*)	Bare coordination with EEBs lagging
2018	0.6645	0.3219	0.5378	0.4932	0.5150	*F*(*x*) > *F*(*y*)	Bare coordination with EEBs lagging

## Data Availability

The data presented in this study are available on request from the corresponding author. The data are not publicly available due to data publisher regulations.

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
