# Peer review of "Coupling Coordination Degree between the Socioeconomic and Eco-Environmental Benefits of Koktokay Global Geopark in China"

_ijerph, 2022, doi:10.3390/ijerph19148498_

Round 1
Reviewer 1 Report
General comments.
First, in many ways this paper is a missed opportunity. The authors present a details method and undertake historical analysis of a case study and the tourism economy. At the same time, the reader is left trying to figure out what the main argument or message is. The authors need to identify a 50 word take a way message, and come back to this theme multiple points in their essay. This is focused on methods, but does not really outline what are the major implications and take a way message.
Second, the authors need to need to define the terms they are employing. There is a lot of assumed knowledge here, and the readers are going to stop reading very, very quickly
Third, and related to the above, the authors need to reduce their use of jargon to maximize the readership. For example they need to step back and clearly present/translate their argument or what things means. Line 110 onward is a good example of this. At the moment the readership will not follow the argument or methods. The authors need to slow down, find simple words, and explain their observations in clear language. This is also the same for the figures. Fig. 2 is all about economic viability and the tourism economy, but in many places simple words can be used here.
Finally, I might have missed something here, but there appears to be no consideration of the interests of people (tourists) in this analysis. Heritage and environmental tourism is based on the interests of people, and their experiences. What do people who attend the parks enjoy? what do they want to get out of it? Humans are agents. Other than presenting a historical analysis of human action, there reader is provided with no sense of the drivers of tourism. Even the concept of the Natural Geological Museum (170) is not really defined. So, how is all of this connected to people? This is very important and the authors need to address this.
Specific comments
Line 180. This definition needs to come much earlier. Probably in the first para.
Tourism. Where are they from, what are they doing? What are they thinking?
Terms. Mild disorder, etc. What does this all mean?
Author Response
Dear Editor and Reviewer,
Thank you for your letter and for the reviewers’ comments concerning our manuscript entitled “Measurement of the Coupling Coordination Degree between the Socioeconomic Benefits and Eco-environmental Benefits of the Koktokay Global Geopark in China”. Those comments are all valuable and very helpful for revising and improving our manuscript, as well as the important guiding significance to our researches. We have studied the comments carefully and the manuscript has been revised carefully and strictly according to your comments, which we hope meet with approval.
Thank you for your constructive comments again, in order to facilitate your review, bold and marked red fonts were used to show the corresponding response. And we have also updated our manuscript by properly adding these responses into the revised version.
All my best wishes for the future.
Sincerely

Reviewer 2 Report
The article "Measurement of the Coupling Coordination Degree between the Socio-economic Benefits and Eco-environmental Benefits of the Koktokay Global Geopark in China" explores a case study on water values in documents. This research experiment involves research approaches by applying semantic keyword analysis and participant perception categorization. The approach of water values developed in this study points out how the authors take the motivations of studies involving water, such as business performance or its conservation, health, nutrition, and human development.
Overall, I have no qualms about the methods presented, however, there is room for improvement by expatiating additional linking statements in terms of water valuing research works and the social background trends. Also, please increase the visibility for the potential readers.
Please take English editing service with a professional proofreader and ask her/him to rephrase the sentences both in results and conclusion sections.
I think this paper is almost ready for publication for now.
Author Response
Dear Editor and Reviewer,
Firstly, it is a great honor to receive your recognition concerning our manuscript entitled “Measurement of the Coupling Coordination Degree between the Socioeconomic Benefits and Eco-environmental Benefits of the Koktokay Global Geopark in China”. We have revised the manuscript carefully according to your comments, which we hope meet with approval. Secondly, we take this opportunity to wish you good health and every success in your work. Finally, we look forward to the next kind cooperation with MDPI. Thank you for your recognition again.
Sincerely

Reviewer 3 Report
I appreciate the detailed research you conducted for this paper. This work has good potential. I have several suggestions of ways to improve what has been done here.
There are some global comments that may help the paper that I would like to make:
Definitions: You need to describe what you mean by socio-economy (more commonly “socioeconomic”) and eco-environment. These exact terms are not commonly used in the papers I read. I was able to guess, but you might want to be more explicit in setting up the structure of the paper to summarize the planned definition of these terms as you use them many times. The same thing goes for “geopark”. Besides giving the UNESCO definition you need to distinguish how they are really different from regular parks. Is it only the UNESCO designation? Examples of other “geoparks” might be helpful. The idea of a socio-economic system and an eco-environment system is interesting, and it would help the reader if that was better described in context of other studies. Also, the idea of “coupling coordination degree” is another concept that needs to be better contextualized. Additionally, what do you mean by “dynamic evolution trend”.
Acronyms : You use a lot of these, which I understand, but after awhile it gets to be confusing. You define these at the beginning, but by the end of the paper the reader might need some refreshing. So, I suggest retelling what the acronyms stand for at the beginning of the last two or three paper sections.
Table 1 Indicators: Perhaps you should put the table in later. The description of the indicators and how they were calculated needs more detail. Even with the discussion I am unsure how you derived each indicator even though you talk of it. It would help to get some real data and go through the calculation of a few of the indicators.
Table 2: You need to be more explicit on the meaning of the different categories that you got from Lao in making the table. What do disorder, lagged, and synchronized mean for example? The table is not helpful as is.
Data sources: You have a paragraph on the data sources, but it would be very helpful if you showed examples of the data themselves. Also, how did you standardize all the of the data so you could make the weights comparable across the different systems? Can you share the data links? Once again, showing how you calculated some of the indicators would help the reader understand the form the data are in.
Discussion and Conclusions: You summarize what you did pretty well, and you talk about the issues related to running geoparks at the end that helps the reader see what you believe are the important conclusions of your research. I am wondering if you can really relate or apply your research to other geoparks (your thoughts in the conclusions)? Since you have done one case study, can your conclusions be extrapolated to other places? Perhaps discussing how or why the Koktokay Geopark is similar to or different from other UNESCO parks can help strengthen your recommendations.
I am including the PDF with some of my editing comments. I made a number of comments at the beginning of the paper, but stopped making comments in the middle. There are not many grammar mistakes, which I commend you on, but the paper is difficult to read because of the acronyms and the assumptions it makes about the reader’s knowledge of the field. I recommend that you find another English reader to take the paper and edit it for readability and not just for grammar.

Author Response

(The authors gave the same response as above.)

Round 2
Reviewer 1 Report
With the minor revisions outlined below this paper is ready to be published.
I have three suggestions that will improve this paper and increase the impact of the paper on the readers.
First, and to be honest I am surprised I did not recognize this in my first review, this paper would be improved by 2-3 photographs of the Koktokay Global Geopark. This paper is focused on the interest of the public in landscape and geological heritage (see lines 53-54). The driver of all of this is the landscape setting, and the experinces of the people who travel and visit at these locations. For the readers to understand the draw to these locations they need to see what the setting looks like, what the park looks like, and perhaps some of the facilities. My suggestion here is to add 2-3 color photographs, including a broad landscape setting photograph taken from the air (think here of a drone photograph, or one from a higher location overlooking some of the park), as well as one or more photographs that shows greater detail, perhaps a river valley, a close up of an important geological exposure. The readers need to see landscape setting of Koktokay Global Geopark so they can contextualize it.
Second, there are places where the layout and formatting needs to be addressed
For example, there is a gap with Lines 600-622, and in other cases the layout could be better. This not the fault of the authors and needs to be addressed by the journal.
Third, I would ask the authors to look closely at the figure and table captions and consider if they can be improved and if the figures are doing what the authors want. For example, Fig 1 could be improved, and the caption, currently listed as “location of the study area”, misses the mark. Fig. 1 is actually four figures, equal in size. The core and most important figure is the topographical map in the lower right side with elevation that shows specific landmarks (and note this is not identified in the caption). The other three figures are inset maps that decrease in scale and show the location of the park. An improved figure would have a larger color elevation map (as per lower right expanded to 200% or more), and then a smaller map of China (such as in the upper left) with a dot showing the location of the Koktokay Global Geopark. As an aside, what are the visuals to the right of the map of China. This looks to be a ghost from another publication. The other maps are acceptable for a technical report but don’t add anything to the paper. In fact, they take away from the paper by detracting from the map in the lower right. So, go with the lower right elevation/park map, enlarged, and with the China map as a inset, perhaps at 25% scale. As to the figure captions. While technically correct, a much stronger figure caption for the proposed new map would be “Fig. 1. Map of Koktokay Global Geopark showing geological and heritage resources (or whatever this is showing) and elevation changes, indicating location within Northwestern China.” Do you see how this is a better figure and caption. It is always a good thing to double check and ask if the figures and captions are doing what we want them to do.
Again, this is a solid paper and the authors have done a good job of addressing my concerns.
Author Response
Dear Editor and Reviewer,
Firstly, it is a great honor to receive your recognition concerning our manuscript entitled “Measurement of the Coupling Coordination Degree between the Socioeconomic Benefits and Eco-environmental Benefits of the Koktokay Global Geopark in China”. We have revised the manuscript carefully according to your comments, which we hope meet with approval. In order to facilitate your review, bold and marked red fonts were used to show the corresponding response. And we have also updated our manuscript by properly adding these responses into the revised version. Secondly, we take this opportunity to wish you good health and every success in your work. Finally, we look forward to the next kind cooperation with MDPI. Thank you for your recognition again.
Sincerely

Reviewer 3 Report
Thank you for your efficient and timely efforts to revise the manuscript. It is much improved. Nice work.
At this point I would suggest a thorough rereading by an outside reader to check for inadvertent grammatical errors, which often come up when one revises a paper. I also think that you could divide the final paragraph into two or three paragraphs to make it more readable. This is important since you have added some material there and as it is, it is quite long. If you divide the paragraph you could have a statement or two at the very end to tie the whole paper together in a neat summary.
Author Response

(The authors gave the same response as above.)

Round 3
Reviewer 3 Report
I appreciate the changes you made to the end of the conclusion. They help strengthen the articulation of the contributions of your paper. I didn't see many editing changes for the rest of the paper. The figures 2 photos are helpful and useful as well. I still think that you could do a more thorough reread by a non-author.
Author Response
Dear reviewer,
We would like to thank you for your time and effort in reviewing our paper and providing constructive comments. Those comments are all valuable and very helpful for revising and improving our manuscript, as well as the important guiding significance to our researches. We have studied the comments carefully and the manuscript has been revised carefully and strictly according to your comments, which we hope meet with approval.
We are here resubmitting the revised manuscript entitled “Coupling Coordination Degree between the Socioeconomic and Eco-Environmental Benefits of Koktokay Global Geopark in China” for your kind consideration of its suitability for publication in International Journal of Environmental Research and Public Health.
Our deepest gratitude goes to you for your careful work and thoughtful suggestions that have helped improve this paper substantially.
Best Regards,
Authors
Point 1: I appreciate the changes you made to the end of the conclusion. They help strengthen the articulation of the contributions of your paper. I didn't see many editing changes for the rest of the paper. The figures 2 photos are helpful and useful as well. I still think that you could do a more thorough reread by a non-author.
Response 1: Thank you very much for all your comments and suggestions, which are very helpful for the revision of this paper. In order to improve the readability of the paper, we not only found a non-author to do a more thorough reread, but also invited the Language Editing Service Team (https://www.mdpi.com/authors/english) to check the language of the whole paper again. Besides, in order to facilitate your review, marked red fonts were used to show the corresponding modifications. And we have also updated our manuscript and added the certificate screenshots of the Language Editing Service this time accordingly. We hope to meet your requirements.
We tried our best to improve the manuscript and made some changes in the manuscript. We appreciate for Reviewer, warm work earnestly, and hope that the correction will meet with approval. Once again, thank you very much for your comments and suggestions.
